# Asymmetrically Substituted Phospholes as Ligands for Coinage Metal Complexes

**DOI:** 10.3390/molecules27113368

**Published:** 2022-05-24

**Authors:** Fabian Roesler, Clemens Bruhn, Rudolf Pietschnig

**Affiliations:** Institute of Chemistry and Center for Interdisciplinary Nanostructure Science and Technology (CINSaT), University of Kassel, Heinrich-Plett-Straße 40, 34132 Kassel, Germany; fabian.roesler@uni-kassel.de (F.R.); bruhn@uni-kassel.de (C.B.)

**Keywords:** organophosphorus chemistry, luminescence, coinage metal, coordination chemistry, chirality

## Abstract

A series of coinage metal complexes asymmetrically substituted 2,5-diaryl phosphole ligands is reported. Structure, identity, and purity of all obtained complexes were corroborated with state-of-the-art techniques (multinuclear NMR, mass spectrometry, elemental analysis, X-ray diffraction) in solution and solid state. All complexes obtained feature luminescence in solution as well as in the solid state. Additionally, DOSY-MW NMR estimation experiments were performed to achieve information about the aggregation behavior of the complexes in solution allowing a direct comparison with their structures observed in the solid state.

## 1. Introduction

Within the diverse group of organophosphorus compounds phospholes play a prominent role [1,2]. Owing to their low HOMO-LUMO gap these five membered heterocycles show pronounced luminescence in form of fluorescence making them excellent materials for a wide range of applications [3,4,5,6,7,8,9]. Recently, we explored the usage of phospholes as guest component in nanoimprint lithography and for the preparation of phosphole-silica hybrids [10,11].

Depending on their substitution pattern, the absorption and emission maxima can be tuned to lower or higher wavelengths and the quantum yield for emissive processes can be controlled to a certain extent [12,13,14,15,16]. In particular, asymmetrically substituted phospholes allow an even more precise adaption of the optical properties [10,17]. Moreover, these compounds have a P stereogenic center, which opens further opportunities [18,19,20]. Straightforward access to asymmetrically substituted phospholes is given via a method originally reported by Märkl et al. [21], for which an improved protocol and mechanistic insights were developed by us recently [13,15,17]. Since the barrier for planar inversion at phosphorus is low, [18] conservation of the stereochemical integrity at the P-stereogenic center is possible by employing the lone pair at phosphorus for derivatization for instance via complexation. In this publication we explore the coordination behavior of an unsymmetrically substituted phosphole with a series of diamagnetic coinage metal centers aiming at P-chiral complexes.

## 2. Results and Discussion

The previously reported β-silyl phosphole **1** has been chosen as an entry for our investigation as it is available in reasonable quantities featuring attractive luminescence properties [10]. Besides its unsymmetric substitution pattern, potential involvement of the thienyl unit as secondary donor site may allow for a variety of coordination motifs in metal complexes of this phosphole.

Removal of the trimethylsilyl (TMS) group furnishes the corresponding β–H phosphole **2** featuring higher quantum yield and unperturbed coplanarity of the adjacent rings with the central phosphole unit [13]. In order to improve the desilylation reaction reported to proceed with KOTMS, we employed tetra-*n*-butylammonium fluoride (TBAF) which is also a well-known desilylation reagent [22,23]. This adaption resulted in a more selective reaction behavior with higher yields. Figure 1 visualizes the corresponding reaction.

With compound **2** in hand, complexation reactions with a series of diamagnetic coinage metal salts were investigated. To this end phosphole **2** was reacted with [Cu(NCMe)_4_]BF_4_, AgBF_4_ and Au(tht)Cl (tht: tetrahydrothiophene) in solution at room temperature. Recorded ^31^P–NMR spectra show a full conversion for all complexation reactions. The ^31^P NMR spectrum of copper complex **3** shows a broad resonance at 5.3 ppm (*Δ*$v_{1/2}$ = 270 Hz), which differs only marginally from the starting material. The signal for the silver complex **4** is shifted by almost 7 ppm to lower field. The most obvious change was found for gold complex **5**, which shows a resonance at 32.5 ppm. These changes are comparable with other phosphorus containing coinage metal complexes [24,25,26,27,28].

For complex **4,** the reaction was carried out in three different solvents (dichloromethane (DCM), acetonitrile (MeCN), and tetrahydrofuran (THF)) affecting line shape and coordination shift (Figure 1). All NMR spectra show a signal in the range around 12 ppm. In DCM two signals are recognizable, while for reaction in the other two solvents only one singlet is visible. Possibly the two signals for the reaction in DCM could arise from two different species or coupling of the observed ^31^P nucleus with the isotopes of silver (^107^Ag and ^109^Ag) both have nuclear spin of ½, which for single coordination would ensue a multiplicity of a doublet [29].

Deconvolution analysis gives a ratio of 46:54 for the two signals, corroborating this latter explanation, since the ratio of the two isotopes is about 48:52.

Since the resonances in DCM are not baseline resolved, no exact determination of the full width at half maximum is possible. The other two signals show a full width at half maximum of *Δ*$v_{1/2}$ = 231 Hz and *Δ*$v_{1/2}$ = 293 Hz. Integration of the corresponding ^1^H NMR spectra revealed that solvent used for the reaction is present in the complex (one equivalent of acetonitrile, vs. half an equivalent of THF) indicating that in these cases there is a coordination of the respective solvents to the silver atom.

### 2.1. Characterization in the Solid State

All complexes were also obtained as single crystals which were analyzed by X-ray structure analysis. Figure 2 shows the molecular structure of copper complex **3**. In complex **3**, two of the acetonitrile units have been removed from the coordination sphere of the copper atom and were replaced by phosphole ligands, which adopt homochiral configurations at the P-stereogenic centers. The copper atom is surrounded in distorted pseudo tetrahedral fashion which is a common coordination motif in the literature [27] contrasting motifs with Cu---Cu interaction which have also been observed with phosphole ligands [28]. The copper-phosphorus distance in **3** is with 2.263(1) Å in the usual range for copper phosphane complexes [30]. The angles between the phosphole ring and the substituents at 2- and 5-position are (2.9(2)° (phenyl) and 13.1(2)° (thienyl)). It is also evident that the sulfur atom of the thienyl ring does not coordinate to the copper center. According to the HSAB principle, however, this would be quite conceivable, as interactions between copper and sulfur occur frequently and even when unintended (e.g., catalyst poisoning) [31]. To gather additional information about the structure of complex **3** in CD_2_Cl_2_ solution, a ^1^H DOSY–MW NMR estimation experiment was performed (Appendix A). The results indicate that in contrast to the composition in the solid state just one phosphole ligand is binding to the copper center in solution. So, in essence, the structure of complex **3** differs in solutions from that in the solid state.

In contrast to copper complex **3**, its silver analog **4**, shows a coordination of the tetrafluoroborate anion to the silver center. The distance is 2.388(3) Å ((Ag1)–(F1)) and within the sum of the van der Waals radii [32]. Furthermore, there is another contact (Ag1)–(F4) with 2.607(3) Å. This results in a polymeric structure in the solid, which is shown in Figure 3. Within the underlying repeating unit, the P-stereogenic centers adopt configurations of opposite chirality. In complex **4**, a silver cation is coordinated by two tetrafluoroborate anions, a phosphorus atom of a phosphole unit ((P1)–(Ag1) 2.373(1) Å) and the sulfur atom (S1A) from a thienyl ring of another phosphole molecule ((Ag1)–(S1A) 2.581(7) Å). The silver center is surrounded in distorted pseudo tetrahedral fashion in this coordination polymer. The angles range from 79.4 to 140.0°, deviating significantly from those of an ideal tetrahedron. The boron-fluorine distances are between 1.371 and 1.398 Å. The larger distances are observed for the coordinating fluorine atoms (F1) and (F4).

In analogy to complex **3**, a ^1^H DOSY–MW NMR estimation experiment was also carried out (Appendix A). It shows that similar to **3** also in **4** only one phosphole ligand is binding to the silver cation in CDCl_3_ solution. The gold complex **5** shows a linear coordination of the gold atom to the phosphorus center (Figure 4).

The Au(1)–P(1) distance is 2.196(2) Å and comparable to related literature values [28,33]. The shortest intermolecular distance between two gold(I) ions is 4.259(1) Å, excluding the presence of aurophilic interactions in **5**. The adjacent aromatic rings in α-position adopt a nearly coplanar arrangement (0.6(4)° (phenyl) and 4.4(4)°(thienyl)) similar to noncoordinated β–H phospholes [13,28]. The performed DOSY–MW NMR estimation experiment shows that in this case the composition in the solid state and in solution are similar (Appendix A). So, in both states the metal center is coordinated by one phosphole ligand containing a center of chirality at phosphorus.

### 2.2. Luminescence Properties

To explore their luminescence properties the absorption and emission maxima in solution (DCM) and solid state have been investigated (Figure 5) for phosphole complexes **3**–**5** and Table 1 summarizes the corresponding data.

Phosphole **2** shows an absorption maximum at 395 nm and an emission maximum at 487 nm in solution. All complexes feature a red-shifted emission and absorption with lower quantum yield referring to the uncoordinated phosphole. The highest quantum yield was found for complex **4** which is still 11% lower than the quantum yield of the free phosphole ligand. The attenuation coefficients are comparable to those obtained for other phospholes [12,13].

In the solid state the situation is more complex. Here no clear tendencies in absorption and emission maxima are observable. But compared to copper and silver complexes of phospholanes, the saturated derivatives of phospholes, the emissions are all strongly blue-shifted [34,35]. Nevertheless, the quantum yields are all lower than those of the phosphole used for the complexation reaction. Polymeric silver complex **4** shows a broader absorption curve than compounds **3** and **5**. The related gold complex of the benchmark triphenylphosphole shows similar emission properties and quantum yields. Unfortunately, the absorption spectrum in solid state was not reported [36].

## 3. Materials and Methods

All reactions were carried out by means of standard Schlenk or glovebox techniques under inert gas atmosphere (argon). Solvents were dried over Na/K alloy before use and were freshly distilled under inert gas.

The used phosphole **1 [10]** as well the corresponding educts phenylphosphane (PhPH_2_) and the unsymmetrically substituted diyne [37] were synthesized as described in the literature.

Deuterated solvents for NMR spectroscopy were dried and stored over molecular sieves. All chemicals were purchased from Sigma-Aldrich (St. Louis, MO, USA), ABCR (Karlsruhe, Germany) or TCI (Tokyo, Japan) and used without further purification.

For purification via column chromatography a puriFlash XS 520 plus (Interchim, Montluçon, France) was used. The used cartridges were filled with spheric silica gel (particle size: 25 µm).

^1^H, ^13^C, ^31^P, and ^29^Si NMR data were recorded on Varian VNMRS-500 MHz or MR-400 MHz spectrometers (CA, USA) at 25 °C. Chemical shifts were referenced to residual protic impurities in the solvent (^1^H) or the deuterated solvent (^13^C) and reported relative to external SiMe_4_ (^1^H, ^13^C, ^29^Si), H_3_PO_4_ (^31^P).

Solution-state structure elucidation of complexes **3**–**5** was performed via a ^1^H–DOSY external calibration curve (ECC) molecular weight (MW) estimation [38,39,40,41] (see Appendix A). Previous studies showed that for most organometallic compounds the dissipated spheres and ellipsoids (DSE) calibration curve is most suitable for an accurate estimation [42]. DOSY–NMR experiments were recorded on a Varian 400 MHz spectrometer. Sample spinning was deactivated during the measurements, and the temperature was set and controlled at 298 K. All DOSY experiments were performed using the Dbppste pulse sequence [43]. Molecular weight estimation was carried out with the software (v1.3) provided by Bachmann [40].

APCI mass determinations were performed on a Finnigan LCQ Deca (ThermoQuest, San Jose, CA, USA). Mass calibration was carried out immediately before sample measurement on sodium formate clusters or by the ESI-Tune Mix standard (Agilent, Santa Clara, CA, USA).

Elemental analyses were performed with a HEKAtech Euro EA CHNS elemental analyzer. Samples were prepared in a Sn cup and analyzed with added V_2_O_5_.

Absorption spectra were recorded using a Shimadzu UV−1900 spectrometer in solution. Emission spectra as well as luminescence quantum yields (absolute method) were measured with the Hamamatsu C11347 system in solution and solid state. For the refinement of the data OriginPro was used. Crystallographic measurements were carried out on a *Stoe* IPDS2 diffractometer with a STOE image plate detector and a Mo-Kα (λ = 0.71073 Å) monochromator or a *Stoe* StadiVari diffractometer with a Pilatus 200K image plate detector and Cu-Kα (λ = 1.54186 Å) radiation. Direct methods were used to solve the measurements and refined by “least-square” cycles (SHELXL−2017) [44]. All nonhydrogen atoms were anisotropically refined without restriction. The evaluation of the data sets and the graphical preparation of the structures were carried out using Olex2 [45] and Mercury [46]. Details of the structure determinations and refinement are summarized in Appendix A. The CCDC depositions 2167961-2167963 contain the supplementary crystallographic data for this paper, which can be obtained free of charge via emailing data_request@ccdc.cam.ac.uk, or by contacting The Cambridge Crystallographic Data Centre at 12 Union Road, Cambridge CB2 1EZ, UK; fax: +44 1223 336033.

### 3.1. Synthesis of Phosphole ***2***

Phosphole **1** (781 mg, 2.0 mmol, 1.0 equiv) was dissolved in THF (10 mL) and a solution of TBAF in THF (1 M, 2.0 mL, 2.0 mmol, 1.0 equiv) was added. The reaction mixture was stirred for 16 h at room temperature. Then, the solvent was removed under reduced pressure and the crude product was purified by flash chromatography (pentane:DCM, 80:20). Phosphole **2** was obtained as yellow, luminescent solid. Yield: 82% (521 mg).

### 3.2. Synthesis of Copper Complex ***3***

Phosphole **2** (32 mg, 0.10 mmol, 1.0 equiv) was dissolved in DCM (3 mL) and added to a solution of [Cu(MeCN)_4_]BF_4_ (16 mg, 0.05 mmol, 0.5 equiv) in DCM (3 mL). The reaction mixture was stirred at room temperature for 90 min. After that the solvent was removed under reduced pressure and the remaining yellow solid was washed with pentane. The product was obtained as yellow, luminescent solid. Yield: 70% (32 mg).

^1^H NMR (399.87 MHz, CD_2_Cl_2_, δ) 7.54–7.09 (m, 12 H, 12 × C*H*_Ar_), 6.94–6.89 (m, 1 H, C*H*_Ar_), 6.76–6.74 (m, 2 H, 2 × C*H*_Ar_), 2.00 (s, 3 H, NCC*H*_3_). ^13^C{^1^H} NMR (100.56 MHz, CD_2_Cl_2_, δ) 144.3 (d, *J*(^13^C–^31^P) = 31.8 Hz, *C*_Ar_), 140.2 (d, *J*(^13^C–^31^P) = 32.7 Hz, *C*_Ar_), 138.6 (d, *J*(^13^C–^31^P) = 18.9 Hz, *C*_Ar_), 134.9 (d, *J*(^13^C–^31^P) = 10.6 Hz, *C*_Ar_), 134.4 (d, *J*(^13^C–^31^P) = 14.2 Hz, *C*_Ar_), 134.0 (d, *J*(^13^C–^31^P) = 15.1 Hz, *C*_Ar_), 133.5 (d, *J*(^13^C–^31^P) = 10.6 Hz, *C*_Ar_), 132.2 (s, *C*_Ar_), 130.1 (d, *J*(^13^C–^31^P) = 9.4 Hz, *C*_Ar_), 129.2 (d, *J*(^13^C–^31^P) = 1.9 Hz, *C*_Ar_), 128.7 (s, *C*_Ar_), 128.4 (d, *J*(^13^C–^31^P) = 1.8 Hz, *C*_Ar_), 127.2 (d, *J*(^13^C–^31^P) = 30.8 Hz, *C*_Ar_), 126.6 (d, *J*(^13^C–^31^P) = 5.9 Hz, *C*_Ar_), 126.2 (d, *J*(^13^C–^31^P) = 1.8 Hz, *C*_Ar_), 126.1 (d, *J*(^13^C–^31^P) = 4.0 Hz, *C*_Ar_), 118.5 (s, N*C*CH_3_), 2.6 (d, *J*(^13^C–^31^P) = 1.8 Hz, NC*C*H_3_). ^31^P{^1^H} NMR (202.30 MHz, CD_2_Cl_2_, δ) 4.7 (s).

Elemental Analysis (%) calcd for C_44_H_36_BCuF_4_N_2_P_2_S_2_: C 60.80, H 4.17, N 3.22, found: C 60.46, H 4.19, N 3.13. MS (APCI–HR) *m/z*: 380.99226 [**2** + Cu]^+^, calcd: 380.99226.

### 3.3. Synthesis of Silver Complex ***4***

Phosphole **2** (32 mg, 0.10 mmol, 1.0 equiv) was dissolved in MeCN (3 mL) and added to a solution of AgBF_4_ (10 mg, 0.05 mmol, 0.5 equiv) in MeCN (3 mL). The reaction mixture was stirred at room temperature for 90 min. After that the solvent was removed under reduced pressure and the remaining yellow solid was washed with pentane. The product was obtained as yellow, luminescent solid. Yield: 84% (36 mg).

^1^H NMR (399.87 MHz, CD_2_Cl_2_, δ) 7.59–7.14 (m, 13 H, 13 × C*H*_Ar_), 7.02–7.01 (m, 1 H, C*H*_Ar_), 6.87–6.84 (m, 1 H, C*H*_Ar_), 2.06 (s, 3 H, NCC*H*_3_). ^13^C{^1^H} NMR (100.56 MHz, CD_2_Cl_2_, δ) 144.3 (d, *J*(^13^C–^31^P) = 34.3 Hz, *C*_Ar_), 138.9 (d, *J*(^13^C–^31^P) = 34.8 Hz, *C*_Ar_), 137.7 (d, *J*(^13^C–^31^P) = 19.1 Hz, *C*_Ar_), 135.9 (d, *J*(^13^C–^31^P) = 10.7 Hz, *C*_Ar_), 134.4 (d, *J*(^13^C–^31^P) = 15.3 Hz, *C*_Ar_), 134.2 (d, *J*(^13^C–^31^P) = 11.8 Hz, *C*_Ar_), 133.6 (d, *J*(^13^C–^31^P) = 14.1 Hz, *C*_Ar_), 133.0 (s, *C*_Ar_), 130.4 (d, *J*(^13^C–^31^P) = 10.4 Hz, *C*_Ar_), 129.7 (s, *C*_Ar_), 129.3 (s, *C*_Ar_), 129.0 (s, *C*_Ar_), 126.9 (s, *C*_Ar_), 126.7 (d, *J*(^13^C–^31^P) = 6.9 Hz, *C*_Ar_), 126.4 (d, *J*(^13^C–^31^P) = 3.4 Hz, *C*_Ar_), 124.7 (d, *J*(^13^C–^31^P) = 31.2 Hz, *C*_Ar_). ^31^P{^1^H} NMR (202.30 MHz, CD_2_Cl_2_, δ) 12.6 (s).

Elemental Analysis (%) calcd for C_40_H_30_AgBF_4_P_2_S_2_: C 57.79, H 3.64 found: C 57.86, H 3.76. MS (APCI–HR) *m*/*z*: 424.96775 [**2** + Ag]^+^, calcd: 424.96775.

### 3.4. Synthesis of Gold Complex ***5***

Phosphole **2** (64 mg, 0.20 mmol, 1 equiv) was dissolved in THF (3 mL) and added to a solution of Au(tht)Cl_4_ (65 mg, 0.20 mmol, 1 equiv) in THF (3 mL). The reaction mixture was stirred at room temperature for 60 min. After that the solvent and tetrahydrothiophene were removed under reduced pressure. The product was obtained as yellow, luminescent solid. Yield: 85% (94 mg).

^1^H NMR (399.87 MHz, CDCl_3_, δ) 7.78–7.74 (m, 2 H, 2 × C*H*_Ar_), 7.63–7.60 (m, 2 H, 2 × C*H*_Ar_), 7.53–7.10 (m, 10 H, 10 × C*H*_Ar_), 6.94–6.91 (m, 1 H, C*H*_Ar_). ^13^C{^1^H} NMR (100.56 MHz, CDCl_3_, δ) 142.7 (s, *C*_Ar_), 142.1 (s, *C*_Ar_), 139.1 (s, *C*_Ar_), 138.5 (s, *C*_Ar_), 137.5 (d, *J*(^13^C-^31^P) = 16.7 Hz, *C*_Ar_), 136.0 (d, *J*(^13^C–^31^P) = 14.5 Hz, *C*_Ar_), 135.2 (d, *J*(^13^C–^31^P) = 16.7 Hz, *C*_Ar_), 135.0 (d, *J*(^13^C–^31^P) = 2.8 Hz, *C*_Ar_), 134.1 (d, *J*(^13^C–^31^P) = 14.2 Hz, *C*_Ar_), 131.8 (d, *J*(^13^C–^31^P) = 12.6 Hz, *C*_Ar_), 131.1 (s, *C*_Ar_), 130.3 (s, *C*_Ar_), 129.1 (d, *J*(^13^C–^31^P) = 5.9 Hz, *C*_Ar_), 128.9 (s, *C*_Ar_), 128.5 (d, *J*(^13^C–^31^P) = 8.0 Hz, *C*_Ar_), 126.1 (d, *J*(^13^C–^31^P) = 3.3 Hz, *C*_Ar_). ^31^P{^1^H} NMR (202.30 MHz, CDCl_3_, δ) 32.3 (s).

Elemental Analysis (%) calcd for C_20_H_15_AuClPS: C 43.61, H 2.75 found: C 43.71, H 2.89. MS (APCI–HR) *m*/*z*: 515.02921 [**5** – Cl]^+^, calcd: 515.02921.

## 4. Conclusions

In summary, we have synthesized the known asymmetrically substituted phosphole **2** via an improved method and explored its coordination behavior towards a series of diamagnetic coinage metal fragments. As a result, three different coinage metal complexes (**3**–**5**) were obtained and characterized in solution and solid state. While Ag complex **4** has a polymeric solid state structure, its Au analog **5** is monomeric without any aurophilic interactions. The corresponding Cu complex **3** is monomeric as well featuring two phosphole ligands per metal. The complexes are soluble in dichloromethane with coordination shifts of the ^31^P NMR resonance increasing with nuclear charge of the coinage metal. Using ^1^H DOSY–MW NMR experiments we estimated the aggregation of all compounds in solution. As a result, the complexes are monomeric with one phosphole ligand per metal ion in solution in contrast to the corresponding solid state structures featuring monomeric (Au), polymeric (Ag) or twofold (Cu) coordination behavior. Additionally, the luminescence properties of all complexes were investigated in solution and solid state. The results show luminescence for all compounds but with lower quantum yields for the complexes compared with the free ligand.

## Data Availability

The original data presented in this study are available from the authors.

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
