# Peer review of "Asymmetrically Substituted Phospholes as Ligands for Coinage Metal Complexes"

_molecules, 2022, doi:10.3390/molecules27113368_

Round 1

Reviewer 1 Report

In the submitted manuscript, Rudolf Pietschnig et al. reported the synthesis and characterization of a series of coinage metal complexes with asymmetrically substituted 2,5-diaryl phosphole ligands. The luminescence features in solution as well as in the solid state and the aggregation behavior have been systematically investigated. I think these results are interesting and I would recommend the submitted manuscript for publication in Molecules after considering the following points.

  1. In Abstract, the sentence “All obtained complexes were investigated in solution and solid state with state of the art techniques (multinuclear NMR, mass spectrometry, elemental analysis, X-ray diffraction and).” seems incomplete and thus incorrect. The multinuclear NMR, mass spectrometry, elemental analysis, X-ray diffraction techniques are mainly used to characterize the structure of the compounds.
  2. In Scheme 1, the yields should be given for all the synthesized compound. And the 2- and 5-position mentioned in line 91 or α-position in line 143 of phosphole ring should be marked.
  3. In Figure 1, the typeface of the scale and chemical shift values are too small to be clear. They should be enlarged.
  4. In Figure 4 captions, line 216,223,242 and 259, “5” represents the compound 5, so it should be marked in bold.
  5. In Figure 5, authors need to mention the excitation wavelength in figure captions in all the fluorescence spectra.
  6. In Figure S5, the 13C-NMR spectrum is not of sufficient quality.
  7. The checkcif report data of compound 5 shows two Alert level B error, could it be improved through further refinement?
  8. Some sentences are complicated and need to be improved. For example, the last sentence “Additionally, the luminescence properties of all complexes were investigated in solution and solid state corroborating luminescence for all compounds obtained with lower quantum yields for the complexes compared with the free ligand.”

Author Response

We thank the reviewer for the helpful comments.

@1: The sentence has been rephrased to make it more precise plus eliminating the typo.

@2: Yields have been added to scheme and the α- and β- positions have been denoted in the phosphole ring as suggested. In addition, a more detailed scheme has been added to the SI, where the ring numbering in addition to the above mentioned (relative) descriptors  α and β are illustrated.

@3: has been enlarged, as suggested.

@4: text style has been corrected to bold.

@5: Excitation wavelengths have been added.

@6: an expansion of the relevant section has been added for better distinction of the signals. In fact, quality in terms of signal to noise ratio is adequate and all signals are visible, while purity of the compound was demonstrated by elemental analysis.

@7: the respective structure solution has been re-refined with details given in the SI. All related geometric values have been updated, as was the deposition at the CCDC.

@8: we rephrased the section avoiding long sentences to convey the message more clearly.

Reviewer 2 Report

The authors report the synthesis of Cu(I), Ag(I) and Au(I) coinage metal complexes based on asymmetrically substituted 2,5-diaryl phosphole. This is a well-written and executed study worthy of publication. I have a few suggestions/corrections for the authors to address:

  • Line 11, delete “and” or replace.
  • Take into consideration the next papers and review: a) G. B. Boursalian, E. R. Nijboer, R. Dorel, L. Pfeifer, O. Markovitch, A. Blokhuis, B. L. Feringa, J. Am. Chem. Soc. 2020, 142, 16868–16876; b) E. Regulska, C. Romero-Nieto, Materials Today Chemistry 2021, 22,100604; c) A. V. Shamsieva, E. I. Musina, T. P. Gerasimova, R. R. Fayzullin, I. E. Kolesnikov, A. I. Samigullina, S. A. Katsyuba, A. A. Karasik, O. G. Sinyashin, Inorg. Chem. 2019, 58, 7698–7704; d) E. I. Musina, A. V. Shamsieva, I. D. Strelnik, T. P. Gerasimova, D. B. Krivolapov, I. E. Kolesnikov, E. V. Grachova, S. P. Tunik, C. Bannwarth, S. Grimme, S. A. Katsyuba, A. A. Karasik, O. G. Sinyashin, Dalton Trans., 2016, 45, 2250-2260.
  • Line 74, delete one word “explanation”.
  • Line 120, a dot is missing.

Author Response

We thank the reviewer for the helpful comments.

@ comment 1: “and” has been removed and sentence was rephrased.

@ comment 2: the suggested references have been added in their respective contexts as refs 2g, 11b, 22a,b

@ comment 3: has been corrected

@ comment 4: missing dot has been added